# Immune complex disease in a chronic monkey study with a humanised, therapeutic antibody against CCL20 is associated with complement-containing drug aggregates

Susan B. Laffan[1], Andrew S. Thomson[2], Shing Mai[2], Cindy Fishman[1], Takahito Kambara[3¤a], Kiran Nistala[4], James T. Raymond[5], Shugui Chen[2], Thulasi Ramani[6¤b], Laura Pageon[6¤c], Rodd Polsky[2], Mark Watkins[1], Gemma Ottolangui[7], John R. White[2], Curtis Maier[1], Michael Herdman[8], Gerben Bouma[4,8]*

1 In vitro In vivo Translation (IVIVT), R&D, GlaxoSmithKline, Collegeville, Pennsylvania, United States of America, 2 Biopharm Analytical Science, R&D Platform Technology and Science, GlaxoSmithKline, King of Prussia, Pennsylvania, United States of America, 3 Pathology, IVIVT, R&D, GlaxoSmithKline, Collegeville, Pennsylvania, United States of America, 4 Adaptive Immunity Research Unit, GlaxoSmithKline, Stevenage, United Kingdom, 5 Charles River Laboratories, Inc., Frederick, Maryland, United States of America, 6 Envigo CRS, Inc., Princeton, New Jersey, United States of America, 7 Biopharm Molecular Discovery, R&D Platform Technology and Science, GlaxoSmithKline, Stevenage, United Kingdom, 8 Clinical Pharmacology and Experimental Medicine, GlaxoSmithKline, Stevenage, United Kingdom

¤a Current address: Pathology, Preclinical Safety AbbVie, North Chicago, Illinois, United States of America
¤b Current address: Global Early Development, Covance CRS LLC, Somerset, New Jersey, United States of America
¤c Current address: Confluence Pathology, LLC, Oakland, New Jersey, United States of America
* gerben.x.bouma@gsk.com

**Data Availability Statement:** All relevant data are within the manuscript and its Supporting Information files.

## Abstract

Despite the potential for the chemokine class as therapeutic targets in immune mediated disease, success has been limited. Many chemokines can bind to multiple receptors and many receptors have multiple ligands, with few exceptions. One of those exceptions is CCL20, which exclusively pairs to CCR6 and is associated with several immunologic conditions, thus providing a promising therapeutic target. Following successful evaluation in a single dose, first time in human clinical study, GSK3050002—a humanized IgG1 monoclonal antibody against human CCL20—was evaluated in a 26-week cynomolgus monkey toxicology study. A high incidence of unexpected vascular and organ inflammation was observed microscopically, leading to the decision to halt clinical development. Here we report a dose-responsive increase in the incidence and severity of inflammation in multiple organs from monkeys receiving 30 and 300 mg/kg/week by either subcutaneous or intravenous injection. Histomorphological changes resembled an immune complex-mediated pathology, which is often due to formation of anti-drug antibodies in monkeys receiving a human protein therapeutic and thus not predictive of clinical outcome. However, the presentation was atypical in that there was a clear dose response with a very high incidence of inflammation with a low incidence of ADA that did not correlate well individually. Additionally, the immunohistologic presentation was atypical in that the severity and distribution of tissue inflammation was greater than the numbers of associated immune complexes (i.e., granular deposits). An

**Funding:** Funding for this study was provided by GlaxoSmithKline and Morphotek, Inc. (NCT01984047). JR is an employee of Charles River Laboratories and principal investigator of the immunohistochemistry analysis. TR is an employee of Covance CRS, LLC (formerly Envigo CRS) and study director of the 26-week monkey toxicity study. LP was an employee of Envigo CRS for the duration of the study. All other listed authors were employees of GlaxoSmithKline (GSK) during the conduct of the study, hold GSK stock or stock options, and meet the criteria for authorship set forth by the International Committee for Medical Journal Editors. Listed authors from GlaxoSmithKline designed the studies and were involved in data collection and analysis, decision to publish and preparation of the manuscript.

**Competing interests:** Funding for this study was provided by GlaxoSmithKline and Morphotek, Inc. (NCT01984047). JR is an employee of Charles River Laboratories and principal investigator of the immunohistochemistry analysis. TR is an employee of Covance CRS, LLC (formerly Envigo CRS) and study director of the 26-week monkey toxicity study. LP was an employee of Envigo CRS for the duration of the study. All other listed authors were employees of GlaxoSmithKline (GSK) during the conduct of the study, hold GSK stock or stock options, and meet the criteria for authorship set forth by the International Committee for Medical Journal Editors. This does not alter our adherence to PLOS ONE policies on sharing data and materials.

extensive ex vivo analysis of large molecular weight protein complexes in monkey serum from this study and in human serum samples demonstrated a time-dependent aggregation of GSK3050002, that was not predicted by in vitro assays. The aggregates also contained complement components. These findings support the hypothesis that immune complexes of drug aggregates, not necessarily including anti-drug antibodies, can fix complement, accumulate over time, and trigger immune complex disease. A situation which may have increased clinical relevance than typical anti-drug antibody-associated immune complex disease in monkeys administered human antibody proteins.

## Introduction

Chemokines and chemokine receptors drive leukocyte recruitment and play an important role in inflammation and inflammatory diseases. While many chemokines can bind multiple receptors and many receptors bind multiple ligands [1], CCL20 is one of the few chemokines that is an exception to this. CCL20, also known as liver and activation-regulated chemokine (LARC) or macrophage inflammatory protein-3α (MIP-3α) is a member of the chemokine superfamily, a group of low molecular weight secreted proteins that function in leukocyte trafficking, recruiting and recirculation [2, 3]. CCL20 has low homology to other chemokines, is the only known chemokine that binds to CC chemokine receptor 6 (CCR6), and CCR6 is the only known receptor that binds CCL20. CCR6 is expressed on a range of immune cell types including T-helper 17 (Th17) cells, T-regulatory cells, natural killer (NK) T-cells, B-cells, γδ T-cells and dendritic cells [2–6]. Neutralizing the biological activity of CCL20 in pre-clinical models of inflammatory diseases inhibited disease activity, suggesting that CCL20 inhibition can impair the migration of CCR6+ cell types into sites of disease activity, thereby impacting the range of proinflammatory activities of these cells in the disease process and leading to a therapeutic benefit [7–9].

GSK3050002, also known as E6071, was initially developed by Morphotek, KAN Research Institute, Inc. and Eisai Co, Ltd., as a humanized IgG1κ antibody that neutralises CCL20 and was being developed for the treatment of psoriatic arthritis, inflammatory bowel disease and potentially other immune-mediated diseases. GSK3050002 was evaluated in a first time in human single dose escalation clinical trial in which 36 healthy male subjects were exposed to single doses up to 20 mg/kg intravenously (IV). This study assessed safety, tolerability, pharmacokinetics and pharmacodynamics of GSK3050002 in healthy male volunteers. GSK3050002 was well tolerated and shown to have a pharmacodynamic effect by inhibiting migration of CCR6+ cells into an experimental skin blister created with suction. There were no adverse events suggestive of an inflammatory response. An anti-drug antibody (ADA) incidence of 11.1% (4 of 36) was detected in clinical subjects who were treated with GSK3050002. The presence of ADA was not associated with any specific adverse events or laboratory abnormalities or change in pharmacokinetics of GSK3050002 [10].

To support a longer duration clinical trial in patients, GSK3050002 was evaluated in a 26-week toxicology study in cynomolgus macaques. The cynomolgus monkey was chosen as a pharmacologically relevant animal species for nonclinical safety evaluation of GSK3050002. Cynomolgus CCL20 protein has 86.5% amino acid sequence homology to human CCL20, and GSK3050002 has been shown to cross react with cynomolgus CCL20. GSK3050002 has a terminal elimination half-life of 5 to 9 days in cynomolgus monkeys. Two toxicology studies were completed with GSK3050002 in cynomolgus monkeys prior to the 26-week study described

here. GSK3050002 was well tolerated in both these prior studies with mild irritation at the dosing site in some animals noted as the only GSK3050002-related finding. The no observed adverse effect level for 5 weeks via IV in monkeys was 100 mg/kg/week, the highest dose level tested. Intravenous administration of GSK3050002 led to the development of ADA in ~20% of animals, but ADA did not appear to have an impact on safety profile or a consistent impact on GSK3050002 serum levels.

In the current study, monkeys received GSK3050002 once weekly for 26 weeks by IV or SC routes at either 0, 30 or 300 mg/kg/week, with a 12 week off-dose period for a cohort of the animals receiving 30 mg/kg/week. This study describes the unexpected findings of vascular inflammation in multiple organs in the monkeys. Additional investigations were performed to understand the underlying mechanism of the observed pathology and comparisons were drawn between findings in monkeys and analysis of *ex vivo* samples from the first time in human clinical study with GSK3050002.

## Materials and methods

### 26-week monkey toxicity study

Twenty-six male and 26 female cynomolgus monkeys obtained from Covance, Alice, Texas, USA were transferred to the test facility Covance CRS, LLC Somerset (formerly Envigo CRS), Inc., Princeton, NJ, USA. The animals were single- or double-housed in stainless steel cages. Compatible pairs, by sex, were housed together per the specifications of USDA Animal Welfare Act (9 CFR Parts 1, 2 and 3). Pairs remained together for group assignment and animals in the same dose group were housed together. If it was not possible to establish or maintain compatible pairs within a group, then animals were housed individually as needed. Each monkey was offered Certified Primate Diet No. 2050C (Harlan Teklad). All primates were fed twice per day. The total amount of biscuits fed for the day was divided in half (in the morning prior to dosing and again in the afternoon). The amount fed was determined by consulting a Primate Feeding Chart according to the Testing Facility's SOP. Diets were supplemented with fruits and/or vegetables and treats were presented as necessary or appropriate. Food was withheld overnight prior to clinical pathology (including urinalysis) collections and prior to scheduled necropsy. Animals were provided with environmental enhancement in accordance with the Animal Welfare Standards; Final Rule (9 CFR Part 3) effective March 18, 1991, following the Testing Facility Standard Operating Procedures. A general check including availability of food and water and environmental conditions was made near the start and end of each working day. Prior to transfer from the in-house colony, all animals were placed in quarantine and allowed to stabilize for a minimum of 5 weeks upon receipt at the Testing Facility. All procedures were designed to avoid discomfort, distress and pain to the animals. Animals experiencing more than momentary or slight pain or distress due to injury or illness were treated by veterinary staff with approved analgesics or agents to relieve pain. Animal were observed for general condition twice daily (once in the morning and once in the afternoon). Study animals were euthanized by exsanguination after ketamine/xylazine-induced anaesthesia and sodium pentobarbital sedation. All studies were conducted in accordance with the GSK Policy on the Care, Welfare and Treatment of Laboratory Animals and were reviewed by Envigo (now Covance) Animal Care and Use Committee Protocol Review Subcommittee.

GSK3050002 was given to cynomolgus monkeys (6/sex at 30 mg/kg/week or 4/sex at 300 mg/kg/week) once weekly for 26 weeks by subcutaneous (SC) injection to the dorsal surface. GSK3050002 was also given to 2 additional groups of monkeys (6/sex at 30 mg/kg/week or 4/sex at 300 mg/kg/week) once weekly for 26 weeks by slow bolus intravenous (IV) injection to the saphenous or cephalic vein. A single group of monkeys were given the vehicle (6/sex at 0

mg/kg/week) for 26 weeks by both subcutaneous and intravenous injection. At the end of the treatment period, 4 animals/sex/group were euthanized and necropsied. Remaining animals (2/sex in each of the 30 mg/kg/week SC and IV groups) as well as control group were held for a 12-week off-dose period. The following endpoints/parameters were evaluated for all animals: clinical observations including dose site evaluations; body weights; qualitative food consumption; ophthalmoscopic observations; electrocardiographic evaluations; routine clinical pathology including hematology, coagulation, clinical chemistry and urinalysis; serum anti-drug antibodies (ADA); immunophenotyping (circulating T, B and T regulatory cells); necropsy observations including organ weights; and macroscopic and microscopic evaluations (over 60 tissues), including immunohistochemical examination for evidence of immune complex deposits in a subset of animals. Please see Supporting information for more detailed description of the toxicity study.

## Toxicokinetic and ADA evaluation

Serum samples were analyzed for GSK3050002 by using a validated analytical method based on immunocapture and trypsin digest, followed by UHPLC/MS analysis as described previously [10]. For immunogenicity analysis, monkey serum samples collected on Day 183 from all main study animals and during the off-dose period at Days 204, 225, 246, and 267, were analysed for anti-GSK3050002 antibodies by a validated bridging electrochemiluminescent immunoassay (ECLIA) on the Meso Scale Discovery (MSD) platform. The ADA analysis was conducted in two phases. In Phase 1, the results were considered to be potentially impacted by the low level of GSK3050002 drug tolerance (~6 μg/mL) in the ADA method compared to the drug concentrations in the monkey serum samples. Therefore, a second bridging ECLIA, incorporating an acid dissociation sample treatment step, was validated and used for sample analysis on all study animals from Days 183 and 267. The acid dissociation ADA method exhibited increased drug tolerance (~45 μg/mL) compared to the original ADA method. Samples that tested positive for ADA were tested in a confirmatory assay in which 100 μg/mL excess unlabelled GSK3050002 was added to determine competitive inhibition of the signal in the anti-GSK3050002 bridging ECLIA. Samples that were considered positive were then analysed further by serial dilution to obtain a titer value for the ADA response. Titer values are reported as the reciprocal of the last dilution before the signal crossed the cut point. If the titer had not crossed the cut point, then the titer was reported as being greater than the last dilution tested. Positive cut points for the initial ECLIA and the acid dissociation ECLIA were determined based on statistical evaluation of the obtained signal in serum samples from control study monkeys with outliers (signal >3.09 x SD of mean vehicle/naïve animals signal) removed or from naïve monkeys, respectively. The 95th percentile of a non-parametric distributed samples size was set as the cut point (106 and 95 ECL counts for initial and acid dissociation ECLIA, respectively). The specificity for both initial and acid dissociation competition assay was 30.4% and samples that were inhibited >30.4% in the presence of excess unlabelled GSK3050002 were considered positive.

## Immunohistochemistry for immune complex disease

Select tissues (liver, heart, mammary gland with skin, kidney, and/or SC injection sites) from a portion of affected monkeys were evaluated by immunohistochemistry (IHC). Paraffin blocks were shipped to Charles River Laboratories, Inc., Frederick, Maryland for IHC analysis. Selected tissues from 8 monkeys, some that were ADA positive as well as ADA negative across both doses and both routes of administration were used. IHC staining was used to detect human IgG (indicative of test article GSK3050002, a human antibody), endogenous monkey immunoglobulins (IgG and IgM), and complement components C3 (Quidel Corporation) and

sC5b-9 /C9 (Dako). For standard amplification, a direct immunoperoxidase procedure was conducted using monkey-adsorbed, goat anti-human IgG and human-adsorbed, goat anti-monkey IgG detection reagents (Bethyl Laboratories, Montgomery, TX). The higher level of amplification used the same 2 detection reagents, and an indirect immunoperoxidase avidin-biotin complex procedure with a biotinylated secondary donkey anti-goat IgG antibody (Jackson ImmunoResearch, West Grove, PA). The IgM detection reagent was goat anti-human IgM (Bethyl Laboratories, Montgomery, TX). Granular deposits containing human IgG, monkey IgG and IgM, C3, and/or sC5b-9 were considered to be evidence of immune complex deposition as determined by a separate immunohistochemistry feasibility study conducted by Charles River Laboratories, Inc (data not shown).

## Matrix assisted laser desorption/ionization (MALDI)

Two SC paraffin block samples were used for exploratory investigations to try to determine the chemical composition and structure of skin tissue deposits by using MALDI imaging mass spectrometry. See Supporting information for more detailed description of the MALDI method.

## Aggregate potential of GSK3050002

Human serum samples from the clinical trial (ClinTrials.gov identifier NCT01984047) [7] were obtained and used for research in accordance with the terms of the informed consent under an IRB/EC approved protocol (approved by the local Ethics Committee (NHS Health Research Authority NRES Committee South Central—Berkshire B). Samples were pooled from each timepoint from multiple subjects that received GSK3050002 (10 or 20 mg/kg). Samples were not pooled across different timepoints to allow for longitudinal analysis. Serum samples from cynomolgus monkeys given 30 or 300 mg/kg/week IV were also pooled from each timepoint, but not across dose levels, route of administration or time points. GSK3050002 was isolated from pooled serum using an anti-idiotype antibody (in house reagent) coupled to activated agarose resin (Thermo Scientific; 26200) to generate an affinity column (1.5 mg capacity). Uncoupled resin was blocked with 4 ml ethanolamine (1M). The slurry was then placed in a 10x100 mm column for analysis using Fast protein liquid chromatography (FPLC; Akta Explorer). Captured GSK3050002 was washed with 15 ml of PBS, eluted using 0.1M glycine pH 2.7 and neutralised using 1M Tris pH 9. The pooled elution samples were then concentrated twice to 0.5 ml in 15 ml 100,000 MWCO Amicon Ultra centrifugal filters and diluted to 1 mg/ml in PBS. Size variants of GSK3050002 were analysed by size exclusion chromatography (SEC) using two columns (Tandem TSKgel Super SW3000 4.6x300 mm; TOSOH Bioscience) in series at a flow rate of 0.2 ml/min on an Agilent HP1100 HPLC system. About 50 μg of sample was injected and variants expressed as percentage of the total product related peak area.

## Western blot protein analysis of isolated aggregates

GSK3050002 isolated from serum by anti-idiotype affinity column were further fractionated and purified using a Superdex 200 10/300 column on an FPLC. For Western blot analysis, approximately 1 μg of protein was reduced (NuPAGE reducing agent) and denatured (Novex SDS sample buffer). C1q, C3 and IgG were detected using antibodies (Abcam; ab71940, Abcam; ab196639 and JIR 315-165-045, respectively) at 1 μg/ml and standard curves from 1 to 5000 ng for each recombinant protein run on the same SDS PAGE gel.

## Results

### Routine endpoints of toxicity study

The toxicology of GSK3050002 was evaluated in cynomolgus monkeys by weekly IV or SC administration of GSK3050002 at 30 mg/kg or 300 mg/kg weekly dosing for 26 weeks followed by a 12 week off-dose period. GSK3050002 was well tolerated (Supporting information) with no deaths or effects on food consumption, body weight, ophthalmology findings, or organ weights. Infrequent (once or twice) clinical signs, including a combination of decreased activity, drowsy appearance, uncoordinated gait, lethargy, intermittent sternal recumbency, whole body pallor and/or pale gums, occurred in four of six animals given 300 mg/kg/week IV. These clinical signs occurred immediately after dosing or within one hour after dosing. The greatest incidence occurred on Day 1 where two males and one female were affected. In all cases, the observations were no longer evident later the same day or by the following morning. Although these observations were seen at very low incidence, of transient nature and were not duration dependent, they were not observed in control monkeys and were only noted in the high dose IV group and were thus considered test article related.

As summarised in Table 1, routine collection and microscopic analysis of skin revealed vascular inflammation of the dermis and subcutis in the mammary and dorsocaudal (non-injection site) regions, with dose-responsive incidence regardless of route. There were no associated clinical or macroscopic observations. Vascular inflammation in the skin (Fig 1A) occurred in approximately half of monkeys at 30 mg/kg/week (4/8 via SC route, and 3/8 via IV route) and nearly all the monkeys at 300 mg/kg/week (8/8 via SC route, and 7/8 via IV route). Inflammation was composed of lymphocytes, plasma cells, macrophages, neutrophils and eosinophils and affected small to medium-sized arteries and veins in the dermis and subcutis. In addition, at the SC injection sites (IS), dose-responsive inflammation (non-vascular) was present in all SC dose groups, including controls (Table 1). Findings were similar between males and females.

In the liver, minimal to moderate inflammation, comprised primarily of macrophages and neutrophils with fewer lymphocytes, plasma cells and eosinophils (Fig 1B), was present in seven of eight monkeys given 300 mg/kg/week SC, six of eight monkeys given 300 mg/kg/week IV and in one female monkey given 30 mg/kg/week IV. Inflammation was diffusely sinusoidal and surrounding portal vessels. The severity of this change was also highest in monkeys given 300 mg/kg/week SC. In the heart, inflammation was present in the myocardium (Fig 1C) of five monkeys (one male and one female given 300 mg/kg/week SC, one male and two females given 300 mg/kg/week IV and in one female given 30 mg/kg/week IV). This finding was characterized by perivascular and transmural infiltrates of lymphocytes, plasma cells, neutrophils, eosinophils and macrophages affecting the epicardium, myocardium and endocardium of the left and right atria and ventricles and valves. One male given 300 mg/kg/week IV and one female each given 300 mg/kg/week SC or 30 mg/kg/week IV had degenerate cardiac myofibers associated with the inflammation. In affected females given 300 mg/kg/week SC or IV, there was also inflammation of the intramural and/or extramural coronary arteries. Inflammation of the arteries and veins in the adventitia of the thoracic aorta (vaso vasorum) was present in the affected SC female. Heart rate increases of 87 to 167 bpm compared to baseline values were observed during Week 24/25 in this female and were potentially related to the multi-organ vascular inflammation. At 300 mg/kg/week, inflammation of the blood vessels in multiple other tissues (aorta, kidney, choroid plexus of brain, pars nervosa of pituitary, esophagus, stomach, duodenum, jejunum, ileum, colon, pancreas, uterus, ovary, thyroids and/or thymus) were observed in six of eight monkeys via the SC route and seven of eight monkeys via the IV route (Table 1 and Fig 1D).

**Table 1. Summary of key individual toxicokinetic data, anti-drug antibody data and H&E and IHC pathology.**

| | | Males | | | | | Females | | | |
|---|---|---|---|---|---|---|---|---|---|---|
| Group | Animal # | TK | ADA | Key H&E Microscopic Observations[a] | IHC Results | Animal # | TK | ADA | Key H&E Microscopic Observations [a] | IHC Results |
| vehicle | 1360 | -- | -- | IS (0/4 SC; 1/2 IV minimal) | NE | 1860 | -- | -- | -- | NE |
| | 1361 | -- | -- | -- | NE | 1861 | -- | -- | IS (2/4 SC slight—minimal; 0/1 IV[b]) | no granular deposits |
| | 1362 | -- | -- | IS (4/4 SC minimal; 0/2 IV) | no granular deposits | 1862 | -- | -- | -- | NE |
| | 1363 | -- | -- | -- | NE | 1863 | -- | -- | IS (0/4 SC; 1/2 IV minimal) | NE |
| | 1364-R | -- | -- | -- | NE | 1864-R | -- | -- | -- | NE |
| | 1365-R | -- | -- | -- | NE | 1865-R | -- | -- | IS (2/4 SC minimal) | NE |
| 30 mg/kg/week SC | 2360 | -- | -- | -- | NE | 2860 | -- | -- | -- | NE |
| | 2361 | -- | -- | IS (2/4 minimal) vessel (skin) | NE | 2861 | -- | -- | IS (2/4 minimal) vessel (skin) | NE |
| | 2362 | -- | -- | IS (3/4 slight—minimal) | NE | 2862 | -- | -- | IS (1/4 minimal) vessel (skin) | no granular deposits |
| | 2363 | -- | -- | -- | NE | 2863 | ↓ | >80 | IS (4/4 slight—minimal) vessel (skin) | NE |
| | 2364-R | -- | 20 | IS (1/4 minimal) | NE | 2864-R | -- | -- | heart vessel (heart) | NE |
| | 2365-R | -- | -- | IS (2/4 minimal) | NE | 2865-R | -- | -- | -- | NE |
| 300 mg/kg/week SC | 3360 | -- | -- | IS (4/4 moderate—marked) vessel (skin, brain, thyroid) heart liver | granular deposits in IS (mononuclear cells), heart (blood vessels), liver (Kupffer cells); no granular deposits in skin | 3860 | -- | -- | IS (4/4 slight—moderate) vessel (skin, brain) liver | NE |
| | 3361 | -- | -- | IS (4/4 moderate—marked) vessel (skin, brain) liver | NE | 3861 | ↓ | >10⁴ | IS (4/4 moderate—marked) vessel (aorta, colon, duodenum, esophagus, heart, ileum, jejunum, kidney, liver, ovary, pancreas, skin, stomach, mammary gland, thymus, uterus/cervix) heart (+degeneration/necrosis) | granular deposits in heart (mononuclear cells), heart (blood vessels), liver (blood vessels), kidney (blood vessels); no granular deposits in skin |
| | 3362 | -- | -- | IS (4/4 moderate—marked) vessel (skin) liver | NE | 3862 | -- | -- | IS (4/4 slight—moderate) vessel (skin, brain) liver | NE |
| | 3363 | -- | -- | IS (4/4 slight—moderate) vessel (skin, brain) liver | NE | 3863 | -- | -- | IS (2/4 slight—moderate) vessel (skin) | NE |
| 30 mg/kg/week IV | 4360 | -- | -- | -- | NE | 4860 | -- | -- | heart liver | NE |
| | 4361 | -- | -- | IS (1/2 minimal) vessel (skin) | NE | 4861 | -- | -- | vessel (skin) | NE |
| | 4362 | -- | -- | -- | NE | 4862 | ↓ | >10⁴ | IS (1/2 minimal) vessel (skin) | no granular deposits |
| | 4363 | -- | -- | -- | NE | 4863 | -- | -- | IS (2/2 minimal) | NE |
| | 4364-R | ↓ | >10⁴ | -- | NE | 4864-R | -- | 100 | -- | NE |
| | 4365-R | -- | -- | -- | NE | 4865-R | -- | -- | vessel (liver) | NE |
| 300 mg/kg/week IV | 5360 | -- | -- | vessel (skin, brain) liver | NE | 5860 | -- | >80 | vessel (skin, brain, heart, mammary gland, pituitary, thymus, thyroid, uterus/cervix) | granular deposits of mononuclear cells in heart, kidney, liver (Kupffer cells); no granular deposits in skin |
| | 5361 | -- | -- | vessel (skin) | NE | 5861 | -- | -- | vessel (skin, brain, mammary gland) | NE |
| | 5362 | -- | -- | IS (1/2 minimal) vessel (skin, esophagus) | NE | 5862 | -- | -- | Liver | NE |
| | 5363 | -- | -- | vessel (skin, brain, mammary gland) heart (+degeneration/necrosis) liver | granular deposits in heart (mononuclear cells), liver (Kupffer cells); no granular deposits in skin | 5863 | -- | -- | IS (1/2 minimal) vessel (skin, brain) liver kidney | NE |

TK = Toxicokinetics; ADA = anti-drug antibody titres; H&E = hematoxylin and eosin; IHC = immunohistochemistry; IS = injection site (4 SC injection sites, 2 IV injection sites—injection sites for IV recovery animals not investigated); -R (shaded) = off-dose animal; - - = no findings; NE = Not examined.

[a] Tissues with inflammation are listed.

[b] one IV injection site skin tissue missing.

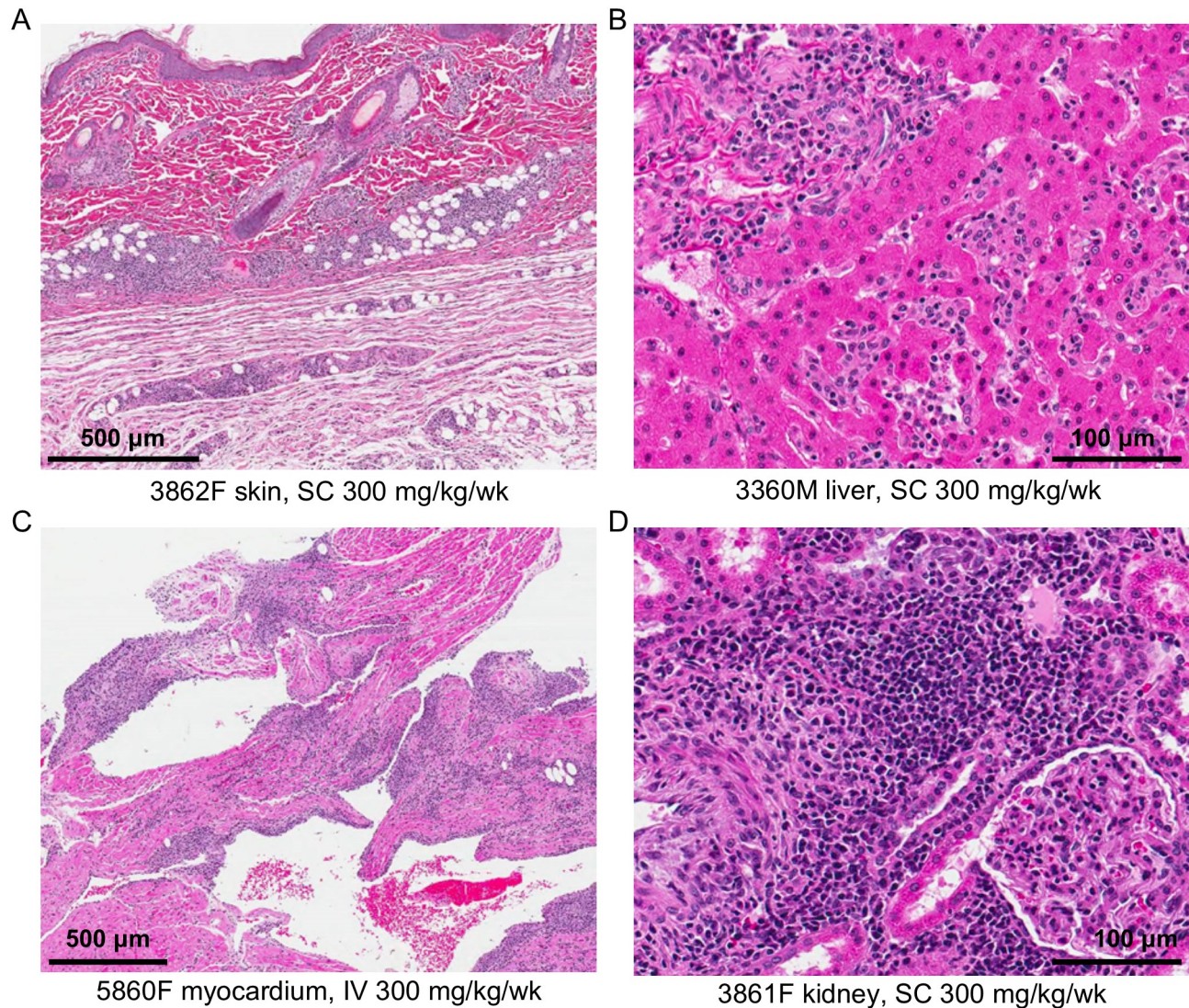

**Fig 1. Inflammation in multiple organs following chronic dosing with GSK3050002.** Representative H&E images of (A) inflammation in skin—non-injection site, (B) sinusoidal and perivascular inflammation in the liver, (C) myocardial inflammation and (D) vascular and perivascular inflammation in the kidney. F = female, M = male, numbers indicate animal identification.

Hematology changes were limited to decreases (to 0.87 times baseline values) in red blood cell mass parameters (red blood cell count, hemoglobin and hematocrit) at Weeks 13 and 26 in individual monkeys given GSK3050002 by either route of administration. One monkey (30 mg/kg/week SC) had increased reticulocyte count, platelet count, red cell distribution width, anisocytosis and hypochromasia consistent with a regenerative response. In addition, one monkey (300 mg/kg/week SC) had low platelets on Week 26 (0.43 times baseline values).

### Reversibility of inflammatory and hematology changes

Following a 12-week off dose period for monkeys previously given 30 mg/kg/week by either the SC or IV route, inflammatory changes seen in the liver, heart, and coronary arteries occurred at a similar incidence and severity to the same findings at the end of the dosing

period, and therefore did not appear to progress or abate (Table 1). Inflammation in the liver was minimally present in one female given 30 mg/kg/week IV. This female had increased liver enzymes at Week 26, which had partially recovered at the end of the 12-week off-dose period. There was no vascular inflammation in the skin and subcutis (non-injections site areas) following the off-dose period which suggests recovery of changes in this tissue for this dose level. The potential for recovery of vascular inflammation in other affected tissues could not be fully evaluated in this study because of the low incidence of these findings in monkeys given 30 mg/kg/week in the main study necropsy (Table 1). At the injection site, following a 12-week off dose period for monkeys previously given 30 mg/kg/week SC, inflammatory changes were now seen at a similar incidence and severity as the same findings in concurrent controls (Table 1), suggesting recovery of test article-related inflammatory changes at SC injection sites for this dose group. There was full recovery of hematologic findings.

### Toxicokinetic profile and anti-drug antibodies

The mean $AUC_{0-168}$ and $C_{max}$ values were similar between males and females at both dose levels and routes of administration for all sampling occasions. Male and female-averaged parameters were therefore used for comparisons (S1 Table). In one male and one female dosed with 30 mg/kg/week IV and one female dosed with 300 mg/kg/week SC, sharply decreased serum concentrations of GSK3050002 were observed after the final dose. Overall lower serum concentrations across the entire profile on Weeks 13 and 26 were also found in one female dosed with 30 mg/kg/week SC. These animals were excluded from calculations of toxicokinetic parameters. Of the eight monkeys in the 12 week off-dose period groups, two males of the 30 mg/kg/week IV group had atypical toxicokinetic profiles with decreased serum levels. The estimated terminal half-life of GSK3050002 based on the toxicokinetic profile of the other six animals ranged from 201 to 266 hours, which is comparable to the previous two toxicology studies (data not shown).

Anti-GSK3050002 antibodies (ADA) were detected with low incidence. As summarised in Table 1, in total, 7 of 40 monkeys receiving GSK3050002 tested positive for ADA (2 of 12 in the 30 mg/kg/week SC, 1 of 8 in the 300 mg/kg/week SC, 3 of 12 in the 30 mg/kg/week IV and 1 of 8 in the 300 mg/kg/week IV groups). Four of these (the same monkeys mentioned in the above paragraph) were associated with decreased GSK3050002 serum levels after repeat dosing, suggestive of immunogenicity reducing GSK3050002 exposure.

### Immunohistochemistry analysis for immune complex disease

Microscopic vascular inflammation appeared consistent with immune complex disease; however, it occurred at a high incidence and without a consistent association with detectable circulating ADA. Therefore, IHC evaluation was performed to determine if immune complex formation and complement activation contributed to the microscopic findings. For this purpose, immunohistologic correlates of systemic and/or localized immune complex deposition that can be disease associated were defined as the presence of immune complex-related granular deposits containing GSK3050002, monkey IgG, monkey IgM, and/or complement components; C3 and/or C9/sC5b-9 [11–13]. IHC was performed on selected tissues from a subset of selected monkeys including animals that were ADA positive as well as animals that were ADA negative across both dose levels and both routes of administration. As summarised in Table 1 (IHC results column), granular deposits could be identified in some animals, both in monkeys that tested ADA positive and ADA negative, but skin was a notable exception. Granular deposits of phagocytic cells and/or blood vessels in the heart, liver and kidney contained monkey IgG, IgM and/or C3, and in some cases also GSK3050002 (phagocytic cells only) (Fig 2). The

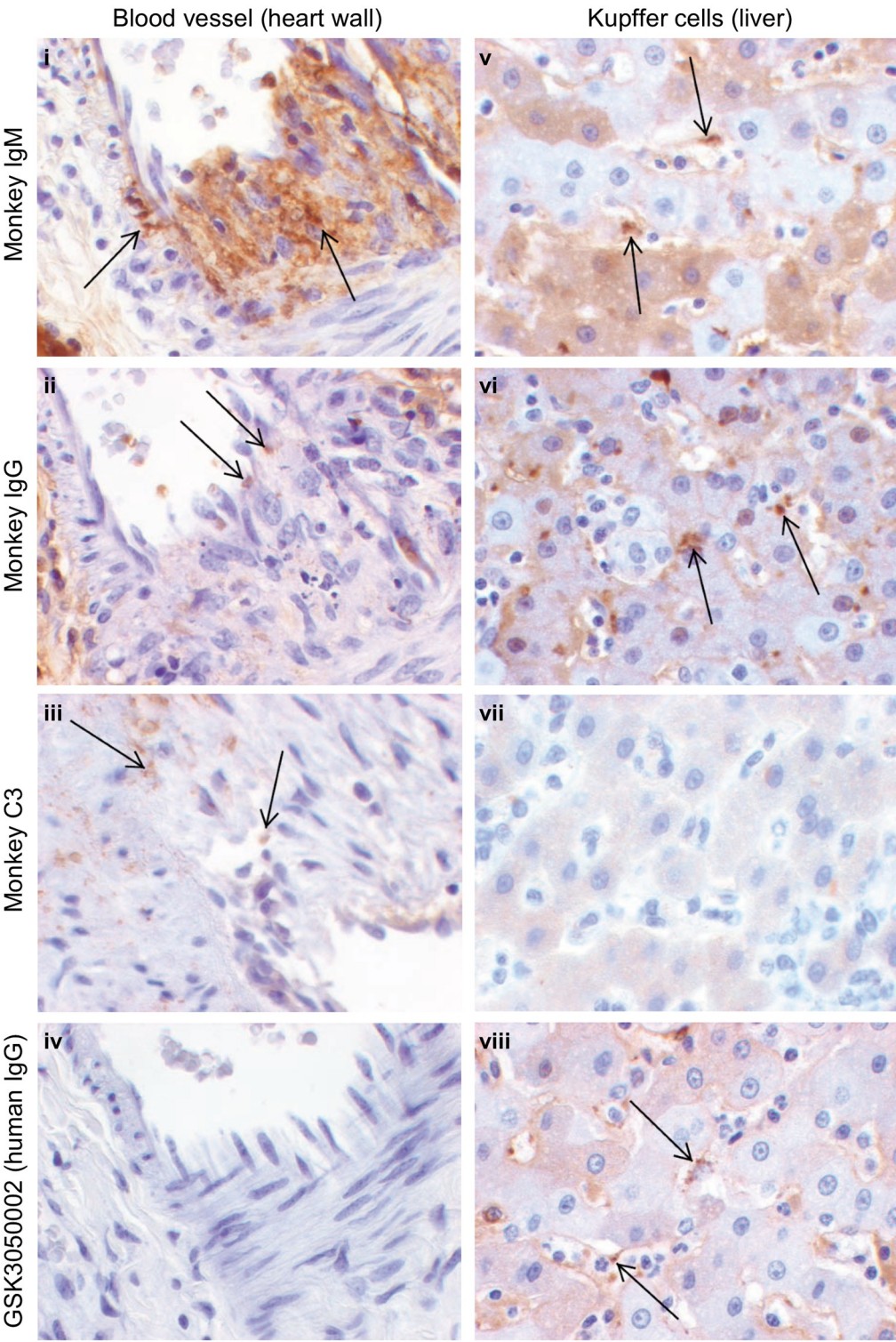

**Fig 2. Immune complex detection following chronic dosing with GSK3050002.** Representative immunohistochemistry images of blood vessels of heart (left panel, i to iii, 40X and iv, 20X magnification) and liver Kupffer cells (right panel, v to viii, 60X magnification) following staining from top to bottom for monkey IgM, IgG, C3 and human IgG (for detection of GSK3050002). Immune complexes, detected as granular deposits (arrows), could be observed for all images except iv and vii. All images from blood vessels (left panel, i to iv) were from Female 3861 (SC 300 mg/kg/week; ADA positive) and images from liver Kupffer cells (right panel, v to viii) were from Male 3360 (SC 300 mg/kg/week; ADA negative).

presence of granular deposits containing GSK3050002, monkey IgG, IgM, and/or C3 was considered consistent with processes of immune complex formation, deposition, pathogenicity and clearance in these animals, although the number of associated granular deposits was considered low for the observed severity and distribution of tissue inflammation. IHC localization of GSK3050002 was detected in skin tissue (extracellular within the dermis including just beneath the epidermis and around hair follicles or in the dermis and subcutis).

## Ex vivo analysis of GSK3050002

In order to determine whether *in vivo* modification of GSK3050002 could be related to the observed monkey toxicology findings, *ex vivo* analysis of GSK3050002 was performed. GSK3050002 was purified from monkey serum from the 26-week toxicology study and compared to serum from healthy human volunteers from the first time in human study [10]. SEC analysis of pooled serum within dose levels, but not across time points, demonstrated the presence of high molecular weight species, suggesting aggregated GSK3050002, which appeared to increase over time in both human and monkey sera (Fig 3A). As expected, normalisation of aggregated GSK3050002 to the dose level received, showed that cynomolgus monkeys that received 300 mg/kg/week had more aggregates than monkeys receiving the 30 mg/kg/week dose (Fig 3B). Furthermore, Western Blot analysis showed the presence of C1q and C3 in the monkey GSK3050002 aggregates (Fig 3C). As C1q requires multimerization of IgG for high affinity binding and subsequent complement activation to occur [14, 15], these findings are supportive of GSK3050002 aggregates providing a matrix for complement activation.

Interestingly, the presence of CCL20 was not detected in the GSK3050002 aggregates by HPLC analysis. As the *in vivo* binding affinity of GSK3050002 for human CCL20 was lower than expected from *in vitro* analysis [10], additional investigations were performed to determine a definitive binding affinity. These investigations confirmed that CCL20 binding occurs at much lower affinity than previously thought, but due to the biphasic nature of the observed binding kinetics, no definitive affinity was obtained either by surface plasmon resonance (SPR) or by solution equilibrium titration (SET) analysis (S1A–S1C Fig). It is plausible that the biphasic binding of GSK3050002 to CCL20 is due to the presence of dimeric CCL20 [16] or the dimeric presentation of the chemokine on tissue surfaces where it is believed to interact with proteoglycans. Analysis of recombinant CCL20 by SDS-PAGE confirmed the existence of a dimer under non-reducing conditions (S1D Fig). As the CCL20 concentration in blood is very low, analysis to determine whether dimeric CCL20 can induce aggregation could not be performed with endogenous CCL20 protein.

## Histologic observations of amorphous-to-crystalline material

Finally, at the SC injection sites, dose-responsive inflammation was present in all SC dose groups, including controls (Table 1 and S2 Table). A dose-responsive increase of intracellular amorphous-to-crystalline material within multinucleate giant cells was observed at inflamed SC injection sites in monkeys in all SC dose groups with one, three and eight monkeys out of eight affected at 0, 30 and 300 mg/kg/week, respectively. Notably amorphous material was also found in a single control monkey at the injection site (S2A, S2B and S2D Fig). In addition, on rare occasions, by the 300 mg/kg/week IV route, multinucleate giant cell formation, sometimes containing amorphous-to-crystalline material, were observed in inflamed areas within the liver, kidney or esophagus in separate monkeys (S2C Fig). The presence of multinucleate giant cells, sometimes containing amorphous material, within inflamed subcutaneous injection sites and tissues is consistent with a foreign body response to particulate material present in, or formed from, the injected formulation. Unsuccessful attempts were made to identify the nature of this

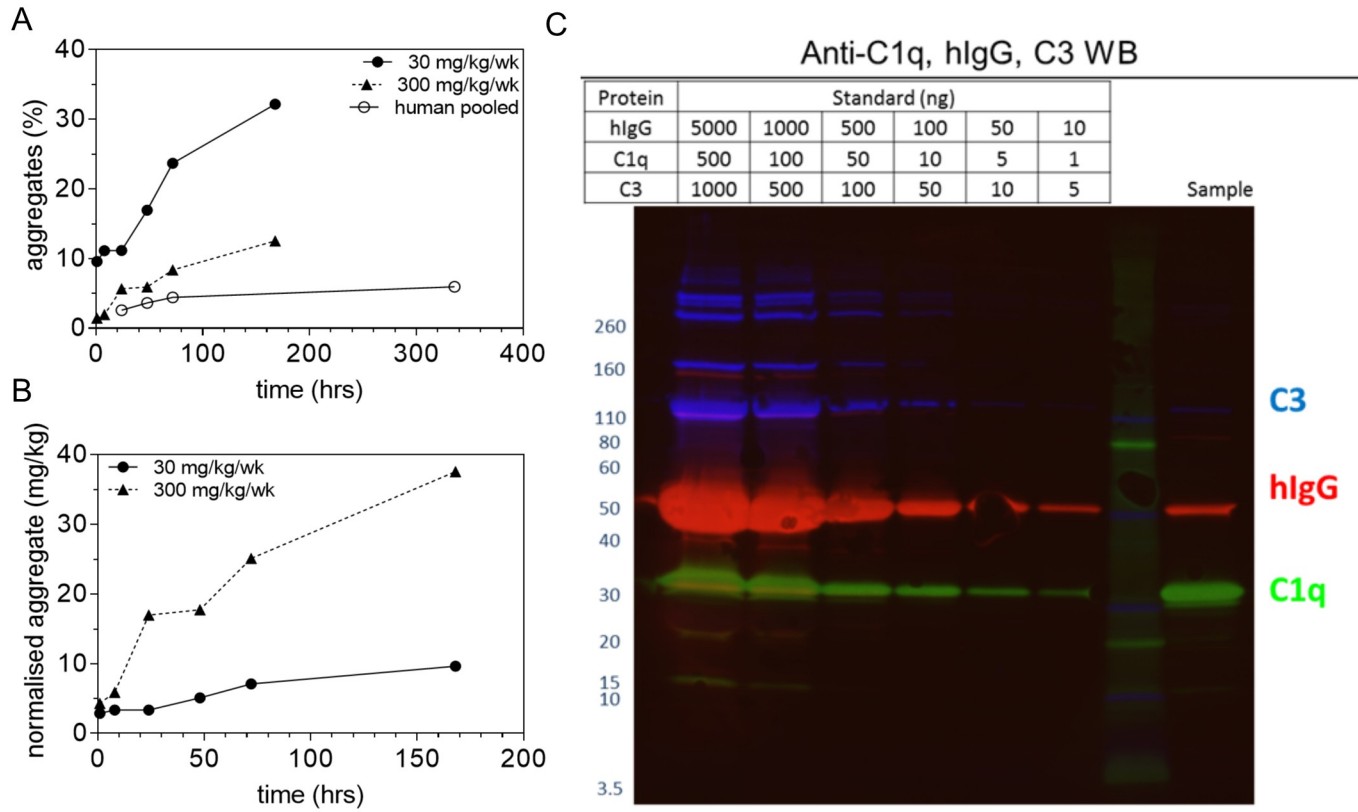

**Fig 3. Aggregation of GSK3050002 and inclusion of complement proteins in aggregates.** SEC analysis of GSK3050002 isolated from sera of human subjects (pooled from 10 and 20 mg/kg doses) and monkeys (30 and 300 mg/kg/week doses) shows aggregated species that increased over ~7 days. (A) Normalisation to dose for monkey serum samples shows higher aggregate level at higher dose level (B). Western blot analysis (C) demonstrates the presence of C3 and C1q in the monkey aggregates.

amorphous material, immunohistochemistry and MALDI analysis were performed. Giant cells stained positive for monkey IgG, IgM and/or C3, but not positive for human IgG (to detect GSK3050002; data not shown). It is also possible that epitopes of GSK3050002 were already masked by endogeneous antibodies for immunohistochemical detection. Furthermore, in a limited set of tissues, MALDI imaging mass spectrometry was performed to identify whether the material consisted of inorganic material or whether it was proteinaceous in origin or contained GSK3050002. However, due to challenging processing steps with formalin-fixed paraffin embedded tissue and the variable and limited presence of the intracellular material in the tissue, no unique signals could be localized to these features and therefore no information regarding their chemical composition could be obtained. Repeated preparation of the drug formulation for injection did not show signs of visible or subvisible particles that were outside the required drug formulation specification. Although the molecular composition of the intracellular amorphous-to-crystalline material could not be identified, it is potentially related to immune complex formation [17, 18] associated with in vivo aggregate formation.

## Discussion

In monkeys, GSK3050002 was well tolerated during treatment, but was found to induce dose-dependent vascular and organ inflammation in several tissues after 26 weeks of treatment. Histological examination resembled immune complex-related pathology. Deposition of immune

complexes within tissues can result in generalised and/or localised pathology, including liver and kidney when immune complexes are cleared from circulation. Vascular inflammation, infusion reactions, glomerulopathies and other effects have been reported to result from immune complexes in monkey toxicity studies with therapeutic humanized antibodies [11, 13, 19–21]. There are examples of biologics with animal findings attributed to immune complexes that have advanced successfully to the clinic and to marketing approval. For example, obinutuzumab, a recombinant humanized monoclonal IgG1 which binds to CD20 expressed on B lymphocytes, had demonstrable immune complex formation and deposition with detectable anti-drug antibodies in some but not all monkeys. Additional findings included arteritis/peri-arteritis and immune complex glomerulonephritis (minimal to marked) as well as fibrosis in the heart noted in some of the high and low dose animals [22].

In this study, inflammation occurred at a high incidence with GSK3050002 and were dose- and duration-dependent, as there was no evidence of this pathology in the 5-week monkey toxicity study. Duration-dependency is consistent with immune complex pathology being more commonly detected in longer term toxicity studies, however, dose-responsiveness and high incidence are not consistent with ADA-associated immune complex disease. Generally, ADA-associated immune complex disease has an idiosyncratic appearance with low incidence and lack of a dose response [23]. Large immune complexes (observed by IHC as granular deposits) were detected in several of the monkey tissues. These immune complexes were found to contain monkey IgM, monkey IgG and/or C3 and in some cases also GSK3050002, although the number of associated granular deposits was considered low for the observed severity and distribution of tissue inflammation.

As a humanized biotherapeutic administered to non-human primates, GSK3050002 has the inherent potential to elicit an immunogenic response. In fact, ADA were detected in some animals, but the incidence was low and the individual correlation with inflammation was poor. The incidence of monkeys with detectable ADA was 17.5% (7 of 40), while the incidence of animals with inflammatory findings, particularly vascular inflammation in the skin, was 63% (9 of 24) in the low dose, and 100% (16 or 16) in the high dose. At an individual level, there was very poor correlation with detectable circulating ADA and inflammatory tissue findings. Some monkeys that were given 300 mg/kg/week showed multi-organ inflammation in the absence of detectable circulating ADAs and the reverse was also found as one monkey that was given 30 mg/kg/week had a high ADA titer in the absence inflammatory findings.

This suggested that the formation of immune complexes of ADA-drug could contribute to the observed pathology but was unlikely to be the sole mechanism of immune complex-mediated pathology. Moreover, evaluation for immune complex formation by IHC in inflamed skin (non-injection site) was negative for granular deposits at all dose levels. Based on this collective information, it is unlikely that ADA is the driving force of immune complex formation and disease. These findings in a weight of evidence along with uncertainty of the mechanism of inflammation, combined with poor monitorability and lack of reversibility, lead to the decision to stop clinical development of GSK3050002.

In a follow-up investigation, *ex vivo* analysis of GSK3050002 after *in vivo* exposure showed that in monkey serum, over 30% of the drug was present in aggregated form and multiple complement proteins could be identified in these aggregates. As expected, monkeys that received the higher dose (300 mg/kg/week) had more aggregates than monkeys receiving the lower dose (30 mg/kg/week), consistent with the observed dose responsive incidence and severity of inflammatory changes. Similarly, although to a lower degree, aggregation was detected in human sera; both monkey groups showed more aggregated drug compared to human samples. Direct comparison between human and monkey sera could not be made as samples from different dose groups of the human study (10 and 20 mg/kg) had to be pooled to obtain sufficient quantity of

GSK3050002 for analysis. No signs or symptoms of vasculitis were observed in the human study (microscopic evaluation of tissues was not performed).

It is plausible that the observed propensity of GSK3050002 to form aggregates containing complement proteins *in vivo* can provide a matrix for complement activation and facilitate immune complex mediated inflammation. Aggregates of exogeneous antibodies are known to cause vasculitis without ADA, since the aggregates can directly fix and activate complement without the participation of endogenous antibodies [21]. In such scenario, the immune complexes do not necessarily contain monkey immunoglobulins (ADA), but rely on the drug to aggregate and activate the classical complement cascade. The high level of C1q in aggregated GSK3050002 could indicate strong complement activation and is supported by the detection of C3 (downstream of C1q) in the aggregates. Multimerization of immunoglobulins into hexamers is required to enable strong binding of C1q and subsequent complement activation [15].

The cause of in vivo GSK3050002 aggregate formation is unknown. CCL20 was found to form weak dimers *in vitro* [16]. If such dimers also occur *in vivo*, this could contribute to drug aggregation. Furthermore, as CCL20 is a chemokine, it may deposit on glycosaminoglycans of the extracellular matrix and endothelium [24]. This allows for a chemokine gradient to form facilitating haptotaxis of CCR6 expressing cells, but it also forms a localised high concentration of CCL20 which could result in high localised levels of GSK3050002, which in turn could contribute to aggregate formation and/or complement activation.CCL20 was, however, not detected in the aggregates, but this may also be a technical limitation of the assay due to elution of the aggregates off the column at low pH. High localised chemokine levels on the endothelium have been described for other chemokines. For instance, in a mouse study, circulating concentrations of CXCL10 increased by 200-fold when CXCL10 was released from endothelial glycosaminoglycan binding following injection of heparin [25]. Moreover, circulating levels of antibody-chemokine complexes can increase to very high levels compared to cytokines and the release of endothelium-bound chemokine by antibodies is thought to play a role in this increase. This was observed in studies with anti-CCL2, where complexed CCL2 levels increased by more than 1,000 times [26, 27]. Similarly, increased GSK3050002:CCL20 levels were observed following dosing with GSK3050002 [10].

CCL20 is expressed constitutively in liver, lung and the appendix and at lower levels in epithelial and endothelial cells in skin, but expression is induced in response to pro-inflammatory cytokines [28]. The expression pattern of CCL20 is consistent with the observed predominant skin histopathology in the cynomolgus monkey toxicology study. Although skin is also a likely target for circulating immune complex deposition [13], granular deposits were not detected in the histology examination. Interestingly, additional analysis of the binding affinity of GSK3050002 to CCL20 revealed that the binding affinity is lower than initially predicted. This is confirmed by clinical data from exposure in healthy volunteers that suggested a binding affinity of approximately 350 pM [10]. It is possible that the low affinity of GSK3050002 and the presence of dimeric CCL20 may contribute to the presence of circulating aggregates if multimeric GSK3050002:CCL20 complexes dissociate from the endothelium. In support of this hypothesis, higher than expected circulating GSK3050002:CCL20 complexes were detected and dissociation of GSK3050002:CCL20 complexes was observed in *ex vivo* clinical samples [10]. Additionally, the low affinity of GSK3050002 may explain why no CCL20 was detected in the circulating aggregates, although this may also be a technical limitation due to elution of the aggregates from the column at low pH. However, mechanisms of aggregate formation independent of target binding cannot be excluded.

It is unclear what the origin is of the observed amorphous to crystalline material, mostly observed in giant cells at the subcutaneous injection sites. There was an increase in amorphous material deposition with increased dose of GSK3050002 to the extent that all monkeys in the

300 mg/kg/week SC dose group were affected. Immunohistochemistry analysis confirmed that the response was associated with inflamed tissue but failed to identify the presence of GSK3050002 in the material. It should be noted that the analyses have limited sensitivity of detection and technical challenges. It is also possible that epitopes of GSK305002 were already masked by endogenous antibodies for immunohistochemical detection. In a further attempt to identify the contents of the amorphous material, MALDI imaging mass spectrometry on FFPE samples was attempted, but methods were not deemed sensitive enough. Furthermore, no obvious deviations were found in the drug formulation process that could be linked to the formation or deposition of amorphous to crystalline material *in vivo*. It appears that the foreign body response with amorphous material required *in vivo* exposure of GSK3050002 for its occurrence, although amorphous material was also found intracellularly at the injection site in one monkey from the vehicle treatment group. It is tempting to speculate that the observed GSK3050002 aggregation *in vivo* plays a role in the foreign body giant cell reaction, but there is no direct evidence to either prove or disprove this hypothesis.

In summary, this report describes findings of inflammation in a 26-week toxicity study in cynomolgus monkeys with a humanized monoclonal antibody against CCL20, GSK3050002, which resembles immune complex disease. However, the presentation was atypical for ADA-associated immune complex disease in that inflammation occurred with a dose response at a high incidence and ADA incidence was low, with very poor individual correlation with inflammation. Follow-up investigations showed that up to 30% of GSK3050002 was present in aggregated form in monkey serum in a dose dependent manner and multiple complement proteins could be identified in these aggregates. Therefore, although ADA may have contributed to pathology in a few monkeys, it is more likely that the circulating drug aggregates were the primary cause of immune complex disease. These findings suggest GSK3050002 may have aggregated to act as a matrix for complement activation, which may have resulted in the observed pathology. Similar drug aggregates, although at reduced levels, were found in samples from the clinical trial with a single dose of GSK3050002 in healthy volunteers. These findings in a weight of evidence including that the lesions cannot be solely attributed to ADA contributed to uncertainty of the mechanism, combined with poor monitorability and lack of reversibility, lead to the decision to halt further clinical development of GSK3050002. In vivo aggregation was unexpected for a therapeutic antibody targeting a soluble target. It is likely that the characteristics of GSK3050002 of reduced affinity, C1q binding and in vivo aggregation propensity played a dominant role in the mechanism leading to the observed immune complex pathology, and these findings may provide important learnings for other therapeutic antibodies.

## Supporting information

**S1 Fig.**
(TIF)

**S2 Fig.**
(TIF)

**S1 Raw Images.**
(PDF)

**S1 File.**
(DOCX)

**S1 Table.**
(DOCX)

**S2 Table.**
(DOCX)

## Acknowledgments

The authors would like to acknowledge the support from the GSK Immunotoxicology Safety Board members (including Daniel Rubio, Maggie Dempster, Eric Lewis) and Antal Rot and Mohamed Daha for assistance with incorporating the toxicology findings, Christopher Carroll, William Bowen and Nadia Tournier for assistance and interpretation of affinity measurements of GSK3050002, Manish Burman for assistance with evaluation of dimeric CCL20 and Steve Castellino for MALDI examination.

## Author Contributions

**Conceptualization:** Susan B. Laffan, Shing Mai, Kiran Nistala, John R. White, Curtis Maier, Michael Herdman, Gerben Bouma.

**Formal analysis:** Cindy Fishman, Takahito Kambara, James T. Raymond, Thulasi Ramani, Laura Pageon, Mark Watkins.

**Investigation:** Susan B. Laffan, Andrew S. Thomson, Shing Mai, Cindy Fishman, Shugui Chen, Rodd Polsky, Gemma Ottolangui.

**Methodology:** Rodd Polsky, Gemma Ottolangui.

**Visualization:** Andrew S. Thomson.

**Writing – original draft:** Susan B. Laffan, Shing Mai, John R. White, Curtis Maier, Michael Herdman, Gerben Bouma.

**Writing – review & editing:** Andrew S. Thomson, Cindy Fishman, Takahito Kambara, Kiran Nistala, James T. Raymond, Mark Watkins, Michael Herdman, Gerben Bouma.

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
