## [Decision Letter · Decision Letter 0]

10 Feb 2020

PONE-D-19-35001

Immune complex disease in a chronic monkey study with a humanised, therapeutic antibody against CCL20 is associated with complement-containing drug aggregates

PLOS ONE

Dear Dr. Bouma,

Thank you for submitting your manuscript to PLOS ONE. After careful consideration, we feel that it has merit but does not fully meet PLOS ONE’s publication criteria as it currently stands. Therefore, we invite you to submit a revised version of the manuscript that addresses the points raised during the review process.

  Please prepare the revised version that includes addressing the comments made both of the reviewers. Reviewer one appeared to be concerned with issues of statistical analyses and data analysis based on sex differences. Reviewer number two while being very positive, had a series of questions and recommendations that I feel important for you and your colleagues to address and incorporate.

We would appreciate receiving your revised manuscript by Mar 26 2020 11:59PM. To enhance the reproducibility of your results, we recommend that if applicable you deposit your laboratory protocols in protocols.io, where a protocol can be assigned its own identifier (DOI) such that it can be cited independently in the future. For instructions see: http://journals.plos.org/plosone/s/submission-guidelines#loc-laboratory-protocols

We look forward to receiving your revised manuscript.

Kind regards,

Aftab A. Ansari, PhD

Academic Editor

PLOS ONE

Journal Requirements:

"All studies were conducted in accordance with the GSK Policy on the Care, Welfare and Treatment of Laboratory Animals and were reviewed the Institutional Animal Care and Use Committee either at GSK or by the ethical review process at the institution

where the work was performed.".   

Please amend your current ethics statement to confirm that your named ethics committee specifically approved this study.

For additional information about PLOS ONE submissions requirements for ethics oversight of animal work, please refer to http://journals.plos.org/plosone/s/submission-guidelines#loc-animal-research  

3. Please note that all PLOS journals ask authors to adhere to our policies for sharing of data and materials: https://journals.plos.org/plosone/s/data-availability. According to PLOS ONE’s Data Availability policy, we require that the minimal dataset underlying results reported in the submission must be made immediately and freely available at the time of publication. As such, please remove any instances of 'unpublished data' or 'data not shown' in your manuscript and replace these with either the relevant data (in the form of additional figures, tables or descriptive text, as appropriate), a citation to where the data can be found, or remove altogether any statements supported by data not presented in the manuscript.

4. In the ethics statement in the manuscript and in the online submission form, please provide additional information about the patient records/samples used in your study (the human samples derived from healthy volunteers enrolled in the clinical trial published here: https://dx.doi.org/10.1111%2Fbcp.13286). Specifically, please ensure that you have discussed whether all data/samples were fully anonymized before you accessed them and/or whether the IRB or ethics committee waived the requirement for informed consent. If patients provided informed written consent to have data/samples from their medical records used in research, please include this information.

5. Please specify whether injections on non-human primates were performed under anaesthesia in order to minimise any pain or distress.

7. Thank you for stating the following financial disclosure:

"YES - Funding for this study was provided by GlaxoSmithKline and Morphotek, Inc. (NCT01984047). JR is an employee of Charles River Laboratories and principal investigator of the immunohistochemistry analysis. TR is an employee of Covance CRS, LLC (formerly Envigo CRS) and study director of the 26-week monkey toxicity study. LP was an employee of Envigo CRS for the duration of the study. All other listed authors were employees of GlaxoSmithKline (GSK) during the conduct of the study, hold GSK stock or stock options, and meet the criteria for authorship set forth by the International Committee for Medical Journal Editors"

8. Thank you for stating the following in the Competing Interests section:

"YES - Funding for this study was provided by GlaxoSmithKline and Morphotek, Inc. (NCT01984047). JR is an employee of Charles River Laboratories and principal investigator of the immunohistochemistry analysis. TR is an employee of Covance CRS, LLC (formerly Envigo CRS) and study director of the 26-week monkey toxicity study. LP was an employee of Envigo CRS for the duration of the study. All other listed authors were employees of GlaxoSmithKline (GSK) during the conduct of the study, hold GSK stock or stock options, and meet the criteria for authorship set forth by the International Committee for Medical Journal Editors"

Reviewers' comments:

Reviewer's Responses to Questions

**Comments to the Author**

1. Is the manuscript technically sound, and do the data support the conclusions?

Reviewer #1: Yes

Reviewer #2: Yes

2. Has the statistical analysis been performed appropriately and rigorously? 

Reviewer #1: No

Reviewer #2: Yes

3. Have the authors made all data underlying the findings in their manuscript fully available?

Reviewer #1: Yes

Reviewer #2: Yes

4. Is the manuscript presented in an intelligible fashion and written in standard English?

Reviewer #1: Yes

Reviewer #2: Yes

5. Review Comments to the Author

Reviewer #1: In this manuscript Laffan et al., conducted very elegant toxicology study involving total of 26 males and female cynomologous monkeys involving GSK3050002 - a humanized IgG1 monoclonal antibody against human CCL20. They found that an dose dependent increase in the incidence of inflammation in several organs/tissues of monkeys receiving low and high does of antibody by subcutaneous or intravenous routes of injection. This is very important finding and warranted for publication after authors address below major and minor issues.

Here is few more major and minor revisions to this manuscript is critical to improve the manuscript quality.

Major: 1) Authors should consultant statistician and needs to perform appropriate statistical measurements in order to say the reported data is statistically significant. 2) Table 1, is not complete in terms of H&E and IHC results, and important to evaluate all animals and to perform statistical analysis for sure about the data -otherwise it will become just observational study.

Minor: 1) Since this manuscript used both sex's of the animals (n=26 per sex) and need to tease out differences if any? and presented any sex differences 2). Please add 1-2 paragraphs to discussion on variety of other therapeutic antibodies with references on the outcome when administered to monkeys with human antibodies.

Reviewer #2: Overall, I found this manuscript to be well organized and the data was presented in a clear way. The text was well-written such that I was able to read it with ease. The information presented seems to me to be of significant value in regard to future studies of therapeutic mAbs that may promote similar types of inflammation.

It is unfortunate that the underlying cause of aggregate formation could not be identified. This could be valuable information. The authors discuss possible ways in which CCL20 might mediate this aggregation, but the failure to detect any CCL20 in the immune complexes does not support this speculation.

I was hoping to learn a little more about the nature of the aggregated GSK3050002 in terms of size. The relative ratio of aggregates was presented in figure 3. I think that it would have been appropriate to present some of the actual SEM data (traces) as well so that the reader could better appreciate the nature of the aggregates. The authors speculate that these aggregates might include hexamers. Wouldn’t that end up in the void for the SEM columns used(?). However, I recognize that characterization of these aggregates is limited by the amounts of aggregate available for analysis.

It was noteworthy that no CCL20 was found associated with the aggregates. I was unclear about the affinity measurements of GSK3050002 for CCL20. The authors seem to indicate that the affinity is much lower than previously believed. This is quite surprising given the fact that this mAb was taken far down the pathway of clinical development. How is it possible that such information was not established prior to the initial human safety trials? Along with the unexpected in vivo aggregation, it is rather amazing that this mAb was developed to the extent that it was.

6. PLOS authors have the option to publish the peer review history of their article (what does this mean?). If published, this will include your full peer review and any attached files.

Reviewer #1: Yes: Siddappa Byrareddy

Reviewer #2: No

---

## [Author Response · Author response to Decision Letter 0]

20 Mar 2020

Editorial comments

1. Please ensure that your manuscript meets PLOS ONE's style requirements, including those for file naming. The PLOS ONE style templates can be found online.

We have re-formatted the manuscript according to the templates provided.

2. Please amend your current ethics statement to confirm that your named ethics committee specifically approved this study.

The ethics statements have been amended to include the requested information and read as follows:

All studies were conducted in accordance with the GSK Policy on the Care, Welfare and Treatment of Laboratory Animals and were reviewed by Envigo (now Covance) Animal Care and Use Committee Protocol Review Subcommittee. 

Human serum samples from the clinical trial (ClinTrials.gov identifier NCT01984047) [7] were obtained and used for research in accordance with the terms of the informed consent under an IRB/EC approved protocol (approved by the local Ethics Committee (NHS Health Research Authority NRES Committee South Central – Berkshire B).

3. Please note that all PLOS journals ask authors to adhere to our policies for sharing of data and materials: https://journals.plos.org/plosone/s/data-availability. According to PLOS ONE’s Data Availability policy, we require that the minimal dataset underlying results reported in the submission must be made immediately and freely available at the time of publication. As such, please remove any instances of 'unpublished data' or 'data not shown' in your manuscript and replace these with either the relevant data (in the form of additional figures, tables or descriptive text, as appropriate), a citation to where the data can be found, or remove altogether any statements supported by data not presented in the manuscript.

Instances of data not shown have been removed and data description removed or included in supporting information.

4. In the ethics statement in the manuscript and in the online submission form, please provide additional information about the patient records/samples used in your study (the human samples derived from healthy volunteers enrolled in the clinical trial published here: https://dx.doi.org/10.1111%2Fbcp.13286). Specifically, please ensure that you have discussed whether all data/samples were fully anonymized before you accessed them and/or whether the IRB or ethics committee waived the requirement for informed consent. If patients provided informed written consent to have data/samples from their medical records used in research, please include this information.

The ethics statement has been updated for clarification. All samples were used in concordance with informed consent and study protocol, which was approved by the local ethical review board. The revised text reads as follows: Human serum samples from the clinical trial (ClinTrials.gov identifier NCT01984047) [7] were obtained and used for research in accordance with the terms of the informed consent under an IRB/EC approved protocol (approved by the local Ethics Committee (NHS Health Research Authority NRES Committee South Central – Berkshire B).

5. Please specify whether injections on non-human primates were performed under anaesthesia in order to minimise any pain or distress.

As per approved protocol, injections were performed without anaesthesia and this information has been added to the methods in supporting information (see lines 26 and 28).

6. PLOS ONE now requires that authors provide the original uncropped and unadjusted images underlying all blot or gel results reported in a submission’s figures or Supporting Information files.

These images have now been provided as S1_raw_images and uploaded as a Supporting Information file.

More detailed information has now been provided as follows: “Listed authors from GlaxoSmithKline designed the studies and were involved in data collection and analysis, decision to publish and preparation of the manuscript. This does not alter our adherence to PLOS ONE policies on sharing data and materials.”

8. Please confirm that the information provided in the Competing Interests section does not alter your adherence to all PLOS ONE policies on sharing data and materials, by including the following statement: "This does not alter our adherence to PLOS ONE policies on sharing data and materials.”

More detailed information has now been provided as follows: “Listed authors from GlaxoSmithKline designed the studies and were involved in data collection and analysis, decision to publish and preparation of the manuscript. This does not alter our adherence to PLOS ONE policies on sharing data and materials.”

Reviewers comments:

Reviewer #1:

Major: 1) Authors should consultant statistician and needs to perform appropriate statistical measurements in order to say the reported data is statistically significant. 

With the exception of the statistical approach to set the cut point for the ADA assay, no other claims are made throughout the paper that the data is statistically significant. As this is a non-human primate study (with a small group size) for which the results of interest for the manuscript were pathologic diagnostic findings made by histology, it would not be appropriate to apply statistical analysis as per consulted statistician advise; thus, no claim of statistical significance has been made. For the main finding (vascular inflammation) the results were in all treated groups of very high incidence and easily interpreted as test article related. It occurred in approximately half of monkeys at 30 mg/kg/week (7/16) and nearly all the monkeys at 300 mg/kg/week (15/16), and not in the vehicle control group. Statistical analysis would not have been informative and is not commonly done is toxicology studies. 

The nature of most other data in this study does not allow statistical analysis as for the clinical sample aggregate analysis, several subject samples were pooled to allow isolation of sufficient amount of aggregates for analysis 

Major 2) Table 1, is not complete in terms of H&E and IHC results, and important to evaluate all animals and to perform statistical analysis for sure about the data -otherwise it will become just observational study.

Apologies for the confusion.For clarity the title of Table 1 has been amended to state that ‘key’ findings are tabulated as it is not practical to put every single finding on the table. Additionally, more information was added to the method section that clarifies that over 60 tissues from each and every animal was assessed (see line 173). These were evaluated by H&E by a board certified veterinary pathologist, and then peer-reviewed by another boarded pathologist. This type of data is not analyzed statistically in general practice. The IHC which was conducted to further characterize the H&E findings for visualization of aggregate deposition in tissues is a highly specialized, high resource effort. Thus, a subset of representative animals and tissues were evaluated. The IHC result is a qualitative analysis conducted by board certified pathologist with specialized training, this type of data is not amenable to statistical analysis. The objectives of the assessment were wholly met, we have taken care in the discussion section to reflect the descriptive nature of the results.

Minor: 1) Since this manuscript used both sex's of the animals (n=26 per sex) and need to tease out differences if any? and presented any sex differences 

Table 1 presents an overview of the key observed findings for individual males and females. A similar frequency of ADA and histology findings was observed in males and females. Grading of histology findings also appears similar between males and females. For added clarity we have added a statement in the results section that findings were similar between males and females (line 284-5). Additionally, lines 335-7 states that toxicokinetic values were similar between males and females at bot dose levels and routes of administration for all sampling occasions.

2). Please add 1-2 paragraphs to discussion on variety of other therapeutic antibodies with references on the outcome when administered to monkeys with human antibodies.

As suggested, we have added detail of toxicology findings related to immune complexes from another therapeutic antibody. A short description has been added to the discussion section (see lines 450-6).

Reviewer #2: Overall, I found this manuscript to be well organized and the data was presented in a clear way. The text was well-written such that I was able to read it with ease. The information presented seems to me to be of significant value in regard to future studies of therapeutic mAbs that may promote similar types of inflammation.

It is unfortunate that the underlying cause of aggregate formation could not be identified. This could be valuable information. The authors discuss possible ways in which CCL20 might mediate this aggregation, but the failure to detect any CCL20 in the immune complexes does not support this speculation.

We have tried to balance the discussion and avoid too much speculation. It is a plausible explanation that CCL20 on endothelium provides a matrix for aggregates to form. The low affinity of GSK3050002 could have contributed to to a release of aggregates from this matrix and could provide an explanation why no CCL20 was detected in the aggregates. Another explanation could be of technical nature that elution from the column at low pH to purify the aggregates has disrupted CCL20:GSK3050002 binding and we have added this to the discussion section (see lines 523-4 and 548-51).

I was hoping to learn a little more about the nature of the aggregated GSK3050002 in terms of size. The relative ratio of aggregates was presented in figure 3. I think that it would have been appropriate to present some of the actual SEM data (traces) as well so that the reader could better appreciate the nature of the aggregates. The authors speculate that these aggregates might include hexamers. Wouldn’t that end up in the void for the SEM columns used(?). However, I recognize that characterization of these aggregates is limited by the amounts of aggregate available for analysis.

The SEC analysis data is described in more detail in a related manuscript. That manuscript describes the comparison of the observed drug aggregates in clinical samples to drug preparations used for in vitro assays. The manuscript has been submitted elsewhere (and was included in the documents provided for original review) and its focus is the detailed description of structural and functional changes of the antibody in different matrices. As the focus of that manuscript is beyond the scope of the manuscript describing the toxicology findings and to avoid publication of the same data, we have not included this in the manuscript. We have removed mention of aggregate size from the discussion section to avoid overlap (see lines 512-4).

It was noteworthy that no CCL20 was found associated with the aggregates. I was unclear about the affinity measurements of GSK3050002 for CCL20. The authors seem to indicate that the affinity is much lower than previously believed. This is quite surprising given the fact that this mAb was taken far down the pathway of clinical development. How is it possible that such information was not established prior to the initial human safety trials? Along with the unexpected in vivo aggregation, it is rather amazing that this mAb was developed to the extent that it was.

The findings that the affinity of our antibody is low, was indeed surprising and came to light following additional analyses after observing unexpected findings of free CCL20 in the FTiH study in which GSK3050002 was well tolerated. Similarly, the findings of in vivo aggregation were unexpected and to our knowledge uncommon for biologics and therapeutic antibodies. Due to the uncommon nature of our findings, we feel this is of interest to the field and we hope that our findings and follow up analyses may provide learnings and benefit development of other therapeutic antibodies.

---

## [Editor Report · Decision Letter 1]

30 Mar 2020

Immune complex disease in a chronic monkey study with a humanised, therapeutic antibody against CCL20 is associated with complement-containing drug aggregates

PONE-D-19-35001R1

Dear Dr. Bouma,

We are pleased to inform you that your manuscript has been judged scientifically suitable for publication and will be formally accepted for publication once it complies with all outstanding technical requirements.

With kind regards,

Aftab A. Ansari, PhD

Academic Editor

PLOS ONE
---

## [Editor Report · Acceptance letter]

1 Apr 2020

PONE-D-19-35001R1 

Immune complex disease in a chronic monkey study with a humanised, therapeutic antibody against CCL20 is associated with complement-containing drug aggregates 

Dear Dr. Bouma:

I am pleased to inform you that your manuscript has been deemed suitable for publication in PLOS ONE. Congratulations! Your manuscript is now with our production department. 

With kind regards,

on behalf of

Dr. Aftab A. Ansari 

Academic Editor

PLOS ONE